# Effect of Low Protein Diets Supplemented with Sodium Butyrate, Medium-Chain Fatty Acids, or n-3 Polyunsaturated Fatty Acids on the Growth Performance, Immune Function, and Microbiome of Weaned Piglets

**DOI:** 10.3390/ijms242417592

**Published:** 2023-12-18

**Authors:** Wenxue Li, Tianyi Lan, Qi Ding, Zhongxiang Ren, Zhiru Tang, Qingsong Tang, Xie Peng, Yetong Xu, Zhihong Sun

**Affiliations:** Laboratory for Bio-Feed and Molecular Nutrition, College of Animal Science and Technology, Southwest University, Chongqing 400715, China; wenxue1998@126.com (W.L.);

**Keywords:** low-protein, sodium butyrate, medium chain fatty acids, polyunsaturated fatty acids, weaned piglets

## Abstract

This study aimed to investigate the effects of low-protein (LP) diets supplemented with sodium butyrate (SB), medium-chain fatty acids (MCT), or n-3 polyunsaturated fatty acids (n-3 PUFA) on the growth performance, immune function, and the microbiome of weaned piglets. A total of 120 healthy weaned piglets ((Landrace × Large White × Duroc); 7.93 ± 0.7 kg initial body weight), were randomly divided into five groups. Each group consisted of six replications with four piglets per replication. Dietary treatments included control diet (CON); LP diet (LP); LP + 0.2% SB diet (LP + SB); LP + 0.2% MCT diet (LP + MCT); and LP + PUFA diet (LP + PUFA). The experimental period lasted for 4 weeks. Compared with the CON diet, LP, LP + SB, LP + MCT, and LP + PUFA diets decreased the final weight and average daily gain (ADG) of piglets (*p* < 0.05). There were lower (*p* < 0.05) concentrations of IL-8 and higher (*p* < 0.05) Glutathione peroxidase (GSH-Px) activity in the plasma of piglets fed with LP + SB, LP + MCT, and LP + PUFA diets than those fed with the LP diet. The piglets in the LP + SB and LP + PUFA groups had lower IKK-alpha (*IKKa*) mRNA expression in the colonic mucosa compared with those in the CON and LP groups (*p* < 0.05). The mRNA expression of *TLR4* in the colonic mucosa of piglets in the LP + SB, LP + MCT, and LP + PUFA groups was decreased when compared with the CON and LP groups (*p* < 0.05). The LP + MCT diets increased the gene expression of nuclear factor erythroid 2-related factor 2 (*Nrf2*) in the colonic mucosa of piglets compared with CON, LP, and LP + SB diets (*p* < 0.05). The abundance of *Erysipelotrichaceae* in the colonic microbiome of piglets in the LP group was higher than that in the other four groups (*p* < 0.05). Collectively, this study showed that LP diets supplemented with SB, MCT, or n-3 PUFA reduced plasma inflammatory factor levels, increased plasma GSH-Px activity, and declined mRNA expression of *TLR4* and *IKKa* in the colonic epithelium, whereas it reduced the abundance of *Erysipelotrichaceae* in the colon of piglets.

## 1. Introduction

The scarcity of protein feed resources, inefficient nitrogen utilization, and environmental pollution issues pose challenges to the livestock and poultry industry. Studies have shown that lower crude protein levels in diets effectively alleviate the nutritional burden of excess protein, diminish hindgut microbial protein fermentation, and reduce nitrogen excretion [1,2]. In particular, one of the important breeders’ aims is to increase growth in farm animals. The enhancement of growth performance is crucial to meet consumers’ demands regarding meat quality [3]. Farm-animal species play crucial roles in satisfying demands for meat on a global scale, and they are genetically being developed to enhance the efficiency of meat production [4]. Moreover, fermentation of undigested proteins and amino acids by intestinal flora is an important cause of post-weaning diarrhea [5]. Therefore, reducing dietary protein levels is a crucial strategy to mitigate diarrhea incidence and improve piglet health [6]. However, significantly lowering crude protein levels in diets leads to reduced growth performance, impaired nitrogen utilization, and compromised intestinal growth and morphology, even with supplementation of essential amino acids deficient in intact proteins or excess free amino acids [7,8]. Hence, it is essential to identify feed additives that enhance nutrient availability in feed to counteract these adverse effects. Recently, studies have reported that SB regulates the energy status of intestinal epithelial cells [9]. SB has been shown to improve piglet immune function through the nuclear factor-kappa B (*NF-κB*) signaling pathway [10].

Furthermore, SB promotes small intestine morphology, enhances beneficial bacteria proliferation, strengthens immunity, and significantly improves piglet growth performance [11]. MCTs are readily absorbed by the intestine and rapidly provide energy, thereby promoting the renewal and repair of intestinal epithelial cells. Additionally, MCT exerts a protective effect on the small intestine and enhances immune function [12]. PUFAs play a vital role in animal growth, development, and immune enhancement. PUFA modulates immune cell activation through the *NF-κB* signaling pathway [13]. It was found that the supplementation of 3% n-3 PUFA to the feed increased the body weight of weaned piglets from 14 d to 28 d [14].

Although studies have investigated the effects of low-protein (LP) diets and SB, MCT, or n-3 PUFA on piglet growth performance, there is limited research on the comparative effects of LP supplementation with SB, MCT, or n-3 PUFA on piglet growth performance separately, plasma immunity, and intestinal health. We speculated that supplementing LP diets with SB, MCT, or n-3 PUFA would have beneficial effects on the growth performance, immune function, and the microbiome of weaned piglets.

## 2. Results

### 2.1. Growth Gerformance and Diarrhea Incidence

As presented in Table 1, compared with the CON diet, LP, LP + SB, LP + MCT, or LP + PUFA diets increased the ADG and ratio of body gain to feed intake of piglets (*p* < 0.05). The ADFI of piglets fed LP, LP + SB, LP + MCT, or LP + PUFA was significantly lower than that of piglets fed the CON diet (*p* < 0.05). As presented in Figure 1, diarrhea incidence and the diarrhea index of piglets in the other groups showed a decreasing trend compared to the CON group (*p* < 0.10).

### 2.2. Plasma Immunity, Inflammatory Factors, and Jejunal Lymphocytes

As shown in Table 2, there were no significant differences in plasma Ig A, Ig G, and Ig M in piglets between the five groups (*p* > 0.05). There was a trend towards lower plasma IL-6 concentrations in piglets fed the LP, LP + SB, LP + MCT, or LP + PUFA diets compared to piglets fed the CON diets (*p* = 0.07). Plasma IL-8 concentrations in the LP + SB, LP + MCT, or LP + PUFA groups were significantly lower than in the LP group (*p* < 0.05). Plasma IL-8 concentrations in the LP + PUFA group were significantly lower than in the CON group (*p* < 0.05). Jejunal lymphocyte counts were higher in piglets fed LP, LP + SB, or LP + PUFA diets than in piglets fed LP + MCT diets (*p* < 0.05).

### 2.3. Anti-Oxidation Capacity

As presented in Table 3, the plasma CAT activity of piglets fed the LP diet showed a tendency to increase compared to piglets fed the LP diet (*p* = 0.10). The GSH-Px activity of piglets was higher in the LP + SB, LP + MCT, or LP + PUFA groups (*p* < 0.05), compared to the CON and LP groups, and the plasma GSH-Px activity of piglets in the LP + MCT group was significantly higher than that in LP + SB group (*p* < 0.05).

### 2.4. The mRNA Expression of TLR4-IKKa and Keap1-Nrf2

As presented in Figure 2. The mRNA expression of *TLR4* in the colonic mucosa of piglets in the LP + SB, LP + MCT, and LP + PUFA groups decreased significantly compared with the CON and LP groups (*p* < 0.05). The LP + SB and LP + PUFA groups decreased significantly the mRNA expression of *IKKa* in the colonic mucosa of piglets (*p* < 0.05), compared to the CON and LP diets; and compared with the CON group, the mRNA expression of *IKKa* in the colonic mucosa of piglets in the LP + MCT group was significantly decreased (*p* < 0.05).

As presented in Figure 3.,there was no significant difference in the mRNA expression of *Keap1* in the colonic mucosa of piglets among the four groups compared with the CON group (*p* > 0.05). The mRNA expression of *Nrf2* in the colonic mucosa of piglets in the LP + MCT group was significantly increased compared with the CON, LP, and LP + SB groups (*p* < 0.05), and the mRNA expression of *Nrf2* in the colonic mucosa of piglets in the LP + PUFA group was significantly higher compared with that in the LP group (*p* < 0.05).

### 2.5. Colonic Microbiome

The number of OUTs obtained was 696 based on nucleotide sequence uniformity between reads (Figure 4A). There were three OTUs specific to the five LP + SB groups, one OTU specific to the LP + MCT group, and one OTU specific to the LP + PUFA group (Figure 4B). The ACE, Chao, Sobs, and Shannon indexes indicated that bacterial community richness and diversity did not differ significantly between treatment groups (Table 4). From the PCA plot (Figure 4C), it can be seen that the species composition of the CON group differed from the LP; LP + PUFA had low similarity with the CON species composition; LP + SB group had high similarity with species composition between CON and LP groups; LP + PUFA group had similarity with all groups. As shown in the PCo plot (Figure 4D), among them, the species composition was similar in the CON, LP, and LP + SB groups, while the species composition was more diverse within the LP + MCT and LP + PUFA groups, especially in the LP + MCT group. The results of the intergroup distance comparison showed that the species composition of the LP + SB and CON groups had high differences. At the family level, *Ruminococcaceae*, *Muribaculaceae*, *Lachnospiraceae*, *Prevotellaceae*, *Lactobacillaceae*, and *Rikenellaceae* were the abundance of *Erysipelotrichaceae,* which was significantly higher in the LP group than in other four groups (*p* < 0.05) (Table 5, Figure 4E). At the genus level, uncultured *bacterium-f-Muribaculaceae* and Lactobacillus were the dominant genera at the level of microbial genera in piglet colonization. Uncultured *bacterium-f-Muribaculaceae*, Lactobacillus (*Lactobacillus, Rikenellaceae-RC9-gut-group*), *Treponema-2*, *Prevotellaceae-NK3B31-group*, Uncultured *bacterium-f-Lachnospiraceae, Faecal Faecalibacterium*, and *Ruminococcaceae-UCG-005* were not significantly different among the five groups (*p* > 0.05); compared with the LP group, *Ruminococcaceae-UCG-014* in the CON, LP + MCT, and LP + PUFA groups had a decreasing trend (*p* = 0.09); *[Eubacterium]-coprostanoligenes*_group in the CON, LP + SB, and LP + PUFA groups compared to the LP + MCT group (*p* = 0.08) (Table 6, Figure 4F).

The marker species with an impact value greater than 3 in the horizontal coordinate were counted by LEfSe analysis (Figure 4G). Three marker species were found in the CON group, *uncultured-bacterium-g-Ruminococcaceae-UCG-013*, *s-Lactobacillus-salivarius*, and *g-Ruminococcaceae-UCG-013*; two marker species were found in the LP + SB group, *s-uncultured-bacterium-g-Peptococcus* and, *g-Peptococcus*; three marker species were found in the LP + PUFA group, uncultured *bacterium-g-Lachnospiraceae-XPB1014-group*, *s-swine-fecal-bacterium-SD-Pec10*, and *g--Lachnospiraceae-XPB1014-group*.

## 3. Discussion

In the present study, we observed that lowering the dietary crude protein level by 3.9 percentage points in the LP group (16.5%) had a detrimental effect on the growth performance of piglets, and the addition of SB, MCT, or n-3 PUFA to the LP diet could not compensate for the effects of LP on growth performance. This result is consistent with previous studies that when feed protein is reduced by ≥3%, numerous amino acids need to be added to the LP ration to meet animal requirements, otherwise growth performance of pigs is reduced [15,16]. Moreover, Wu [17] emphasized that both essential amino acids and nonessential amino acids should be taken into consideration in the formulation of balanced diets to maximize protein accretion in animals. On the other hand, this crude protein level cannot satisfy the growth of piglets due to the poor intestinal development of weaned piglets, which are more sensitive to changes in the protein. In this study, the addition of SB, MCT, and n-3 PUFA continued to fail to compensate for the negative effects of LP on growth performance. The reason could be that the protein was reduced by a larger percentage, thus not meeting the protein requirements for piglet growth. In terms of piglet diarrhea, the diarrhea incidence and diarrhea index of piglets in the LP, LP + SB, LP + MCT, or LP + PUFA groups showed a decreasing trend compared to the CON group. The results of the LP diet to reduce diarrhea in piglets are consistent with the results of a previous study [18]. SB, MCT, and n-3 PUFA also tended to reduce the rate of diarrhea in piglets with some positive effects. It was also demonstrated that the addition of MCFA and SB to premixes of plant origin significantly inhibited enterotoxigenic E. coli diarrhea, thus promoting animal intestinal health [19].

Data from the present study showed that there was a tendency for the LP group to reduce IL-6 compared to the CON group, while there was no further trend of change with the addition of SB, MCT, and n-3 PUFA. Wan et al. [20] reported a reduction in the pro-inflammatory factor IL-6 in LP diets, and previous studies have also suggested a reduction in IL-6 with the addition of SB, MCT, or n-3 PUFA [21]. The experiment results showed that the LP group did not cause a decrease in IL-8 compared to the CON group, and the addition of LP, MCT, and PUFA to LP diets significantly decreased the level of IL-8, especially with the addition of PUFA, which helped to alleviate the inflammatory response of the organism. In agreement with previous studies, feeding LP diets with a protein level of 16% increased IL-8 levels in the serum of 20 kg to 30 kg pigs [22], which may be caused by not enough balanced amino acids. Xu et al. [21] found that oral administration of sodium butyrate to weaned piglets decreased IL-8 expression in the ileum. The regulation of immunity by MCFA is influenced by the type of MCFA of MCFA, in an in vitro study, octanoic acid inhibited IL-8 production by Caco-2 cells, while decanoic acid promoted IL-8 secretion [23]. The MCT used in this study was a mixture of octanoic and decanoic acid, and MCFA was perhaps more effective in immune effects. The results of this study found that there were no significant differences in plasma IL-10 in piglets between the five groups. Inconsistent with previous studies, feeding LP rations significantly increased IL-10 [22]. A related study has found that both SB and n-3 PUFA enhance the production of IL-10 anti-inflammatory factors [21]. The effect of PUFA in promoting IL-10 was more pronounced in the conclusion of the present study. The promotion of IL-10 by LP diets may manifest itself as the experiment of this study continues to be extended. The addition of MCT and n-3 PUFA to LP diets tended to reduce IFN-γ content, with PUFA showing a more pronounced trend, in general agreement with the results of other studies [24]. In general, the addition of SB, MCT, and n-3 PUFA to LP diets has the effect of reducing the inflammatory response.

Weaning stress is a major contributor to oxidative stress in the intestinal tract and blood of piglets [25]. To prevent oxidative stress, antioxidant systems, including antioxidant enzymes of SOD, CAT, and GSH-Px, can reduce the production of free radicals [26]. The results of this study showed that LP diets and LP diets supplemented with LP, MCT, and n-3 PUFA did not affect the plasma levels of T-AOC, T-SOD, TNOS, and iNOS in piglets compared to normal diets; LP diets seemed to affect CAT levels, with a tendency to recover after the addition of SB, MCT, and n-3 PUFA, especially in the LP + MCT group. For GSH-Px levels in this study, the addition of SB, MCT, and n-3 PUFA to LP diets significantly increased the GSH-Px level. Wu et al. found that a 2.2% reduction in crude protein level based on normal rations did not cause changes in T-AOC and CAT content in piglet blood [27]. SB increased CAT and T-SOD content [28]. The possible reason for this result could be that SB was added at an insufficient level to be combined with LP diets, so further research is needed. Capric acid was able to increase T-SOD and GSH-Px mRNA expression in porcine epithelial cells [29]. Perfusion of n-3 PUFA in rats with testicular injury increased the content of T-AOC, CAT, and GSH-Px in tissues, and enhanced the antioxidant capacity of the organism [30]. In summary, the addition of SB, MCT, and n-3 PUFA to LP diets can increase the antioxidant capacity of piglets, especially by increasing the level of GSH-Px in the blood. The detection of the Keap1-Nrf2 signaling pathway at the mRNA level revealed that the mRNA expression of Nrf2 was significantly increased after the addition of MCT and n-3 PUFA to the LP diet compared to the LP group, and the comparison of the results of plasma antioxidant enzyme analysis also revealed that the highest levels of GSH-Px were found in the group with the addition of MCT and n-3 PUFA to the LP diet, which is consistent with the previous results. Therefore, the addition of MCT and n-3 PUFA to LP diets could improve the antioxidant function of weaned piglets.

Higher levels of protein diets increase pro-inflammatory factors and upregulate the expression levels of intestinal *TLR4*, *MyD88*, and *NF-κB* in piglets, leading to a decrease in immune function [31]. *TLR4*, a member of the Toll-like receptor (*TLR*) family, is involved in innate immunity and mediates inflammatory responses by recognizing lipopolysaccharide (LPS) or bacterial endotoxins. Hyperactivation of *TLR4* triggers the production of various inflammatory factors. Under the stimulation of *TNF-α*, *IKKα* can directly enter the nucleus and activate the expression of specific *NF-κB*-responsive genes by catalyzing the phosphorylation of histone H3. It has been shown that adding FA to the diet can reduce inflammation by inhibiting *TLR4* signaling, thereby improving intestinal health [32]. Experimental results showed that LP diets significantly decreased the expression of *TLR4* after the addition of SB, MCT, and n-3 PUFA, and the activation of the *IKKa* signaling pathway activates *IκB* kinase and affects the expression of *NF-κB*. Similarly, we found that the expression of *IKKa* decreased in LP + SB and LP + PUFA compared to the LP group. Related studies also showed that the addition of SB, MCT, and n-3 PUFA had an inhibitory effect on inflammation occurrence [33]. Many studies reported that n-3 PUFA can alleviate the inflammatory response in animals [16]. Researchers have hypothesized that fatty acids of different chain lengths likely attenuate the inflammatory response through the *NF-κB* and *TLR4* signaling pathways [34]. Therefore, the addition of SB, MCT, and n-3 PUFA to LP diets has an inhibitory effect on the activation of colonic inflammatory pathways.

The structural integrity of the gut and the homeostasis of the gut microbiome ensure the chemically induced digestive function of the gut, which is a prerequisite for nutrient absorption, metabolism, and deposition. The intestinal microbiome is considered a key factor in maintaining intestinal function [35]. In the present study, the analysis of colonic microbial diversity revealed that LP diets exhibited OUT specific to the addition of SB, MCT, and n-3 PUFA, with the LP + SB group having the most OUT specific. Alpha diversity analysis also revealed that there was no significant difference between the groups, the LP + SB group had the highest ACE, Chao1, Shannon, and Simpson indexes for each group, and the LP + SB group had a more diverse species composition. It has been confirmed that dietary fiber intake in the diet, or pharmacological treatment of inflammation, which are diet-induced microbial alterations, are accompanied by changes in short-chain fatty acid concentrations [36]. Zhou et al. found no significant effect on Alpha diversity in the study of LP diets [37], which is essentially the same as the results of the present study. Beta diversity analysis by PUG and UPGMA tree analysis showed significant clustering in the CON, LP, and LP + SB groups, suggesting that it affected the microbial community. The reasons for the poor clustering effect exhibited by both LP + MCT and LP + PUFA groups to varying degrees are unclear. Therefore, the above indicates that the addition of SB to LP diets had a more significant effect on microbial diversity.

At the family level, the abundance of the *Daniostoma* family was significantly higher in the LP group than in the other four groups. In recent studies, an association between *Danseraceae* and the development of gastrointestinal inflammation was found, and Chen reported a positive correlation between *Danseraceae* and colon cancer [38]. But a decrease in *Danseraceae* abundance was found in some patients with colon inflammation such as Crohn’s disease [39]. Several studies have also found a decrease in SCFA production and an increase in the abundance of *Danseraceae* under LP ration conditions [40]. Wu’s study indicated that on the prediction of the function of the *Daniostoma* family, it was concluded that the *Daniostoma* family can consume dietary carbohydrates to promote the production of SCFA [41]. Therefore, the increased abundance of *Dantofilaceae* due to LP diets may be related to the deficiency of SCFA production by gut microorganisms, which was alleviated by the addition of SB, MCT, and n-3 PUFA in the gut. At the genus level, the abundance of *Rumenococcus spp.-UCG-014* was higher in the LP group. Sun et al. found that the addition of sodium butyrate to feed reduced the abundance of rumen cocci spp. in the intestine of growing pigs, similar to the results of the present study [42]. *Rumenococcus spp*. functionally affect the conversion of primary bile acids to secondary bile acids in addition to metabolizing some plant fibers [43], and the use of different chain-length fatty acids may affect this process. The LP + MCT group has a more abundant *[Eubacterium]-coprostanoligenes_group* and Koppel reported that this bacterium affects cholesterol digestion and absorption, with nearly half of the cholesterol being excreted directly from the body in the form of fecal sterols [44]. In brief, the results of this experiment suggest that the addition of SB, MCT, and n-3 PUFA suppressed the number of harmful florae, which may be related to the improvement of weaned piglets’ health, intestinal morphology, nutrient absorption, and the reduction of diarrhea in this study.

## 4. Materials and Methods

### 4.1. Experimental Design and Management

A total of 120 healthy 28-day-old weaned piglets ((Landrace × Large White × Duroc); Split evenly between males and females; 7.93 ± 0.7 kg initial body weight) were randomly assigned to one of five diets: 20.4% CP diet (CON); 16.5% CP diet (LP); LP + 0.2% SB diet (LP + SB); LP + 0.2% MCT diet (LP + MCT); LP + 0.2% PUFA diet (LP + PUFA). Each dietary group had six pens with four piglets per pen, two barrows, and two gilts. The experimental period lasted for 4 weeks, and the pigs were transferred to one of five experimental diets over 3 days to gradually reach 100% of the experimental diets on day 4. The composition of the experimental diets (Table 7) met the nutritional requirements of weaned piglets as recommended by NRC (2012). The SB (purity ≥ 98%), MCFA (purity ≥ 50%), or n-3 PUFA (purity ≥ 50%) were sourced from Longyan Xinao Biotechnology Co., Ltd. based in Longyan, China. The molecular formula of SB was C4H7O2Na, then all sodium butyrate. The part of salt was 2%. The main components of MCFA were caprylic and capric acids, with the remaining 50% being medium chain triglycerides. The primary active ingredients of the n-3 PUFA were alpha-linolenic acid, docosahexaenoic acid, and eicosapentaenoic acid. SB, MCFA, and PUFA were protected. The piglets were fed at 8:30, 12:30, and 17:30, and all piglets had available access to feed and water during the study. The environmental conditions were precisely controlled through an automated mechanical system, which kept the ambient temperature within a range of 24 to 28 °C.

### 4.2. Recording and Sample Collection

The weight of the piglets was recorded at 8:00 a.m. on days 1 and 29 of the experiment, and feed consumption was recorded daily. The average daily gain (ADG), average daily feed intake (ADFI), and (G: F; gain to feed ratio) were calculated from days 1 to 28 of the experiment. During the experiment, the experimenter observed clinical signs of diarrhea in weaned piglets daily and used a stool consistency scale to indicate the presence and severity of diarrhea. As shown below: 0 = normal, solid feces; 1 = partially formed soft feces; 2 = semi-liquid feces; and 3 = watery, mucus-like feces. Diarrhea incidence and diarrhea index were calculated according to the following formulae: diarrhea incidence (%) = sum of the number of piglets with diarrhea in each replicate group during the experiment period/(number of piglets in the replicate group × total number of days in the trial) × 100% [45]. Diarrhea index = sum of diarrhea scores in each replicate group during the experiment period/(number of piglets in the replicate group × total number of days in the trial). On the 29th day, six piglets from each treatment group were chosen at random for slaughter. Prior to their slaughter, a blood sample of 10 mL was collected from each piglet through jugular puncture into a sodium heparin-treated 10 mL tube. The plasma was extracted by centrifugation at 3000× *g* for 15 min. Next, piglets were anesthetized with an intravenous injection of sodium pentobarbital (60 mg/kg BW), and the pigs were humanely slaughtered. The piglets were exsanguinated by severing the carotid artery and jugular vein. After washing the colonic tissue with saline, the mucosa was scraped using a surgical scalpel and immediately frozen in liquid nitrogen and then at −80 °C for later analysis. The colonic digesta (approximately 5 g) was collected in sterile tubes for microbiota analysis. The piglet diets (approximately 100 g) were collected.

### 4.3. Chemical Analysis

Plasma immunoglobulins and cytokines were determined by the ELISA kit (Pengguang Biotechnology Co., Ltd. Chongqing, China). Immunoglobulins included immunoglobulin A (Ig A), immunoglobulin G (Ig G), and immunoglobulin M (Ig M). Plasma cytokines included interleukin-6 (IL-6), interleukin-8 (IL-8), interleukin-10 (IL-10), interleukin-17 (IL-17), transforming growth factor-β (TGF-β), and interferon-γ (IFN-γ). The plasma antioxidant capacity was determined using the kit (Nanjing Jiancheng Institute of Biology Nanjing, China), and the procedure was strictly following the instructions for use. The measurement indexes included total antioxidant capacity (T-AOC), catalase (CAT), total superoxide dismutase (T-SOD), glutathione peroxidase (GSH-Px), total nitric oxide synthase (TNOS), and inducible nitric oxide synthase (INOS).

### 4.4. The Gene Expression in Colonic Mucosa

Samples were processed (liquid nitrogen grinding) and total RNA was extracted using Trizol (Code, R0016. Beyotime Biotechnology, Shanghai, China) reagent. The OD260/OD280 ratio of the extracted RNA was between 1.8 and 2.0. The concentration of total RNA was measured using a NanoDrop-ND2000 spectrophotometer (NanoDrop Technologies, Inc., US) and according to the manufacturer’s instructions. Qualified RNA samples were reverse transcribed using Evo M-MLV Reverse Transcription Reagent Premix (Code. AG11706; Accurate Biotechnology Co., Ltd. Hunan, China.). The primer sequences of the target genes were designed and validated in NCBI GenBank (Table 8). Real-time PCR analysis was performed using the SYBR Green method in combination with the ABI 7900 Sequence Detection System. Thermal cycling conditions used for PCR were denaturation at 94 °C for 30 s, annealing at 60 °C for 30 s, and extension at 72 °C for 30 s for 40 cycles. In addition, melting curve analysis was used to ensure that PCR products remained specific and pure. The relative expression ratio (R) of mRNA was calculated as R = 2-ΔΔCt (sample-control) using the Ct method formula for gene expression and comparing the differences [46].

### 4.5. The Microbiome in the Colon

The total genomic DNA was extracted and purified using a DNA Kit (Qiagen, Hilden, Germany) from 0.5–1 g of colonic chyme per sample. DNA integrity was assessed by 1% agarose gel electrophoresis and DNA concentration was measured by NanoDrop 2000 spectrophotometer (NanoDrop Technologies, Inc., Wilmington, DE, USA). The 16S RNA V3–V4 gene region was amplified by using the primers 338F and 806R. The PCR amplification was conducted in a 20 μL reaction system, using the following amplification procedure: initial denaturation at 96 °C for 3 min, followed by 27 cycles consisting of denaturation at 95 °C for 30 s, annealing at 55 °C for 30 s, extension at 72 °C for 45 s, and a final extension at 72 °C for 10 min. The amplified fragments were then subjected to analysis by 2% agar gel electrophoresis. The products were further purified using the AxyPrep DNA Gel Extraction Kit (Axygen Bioscience, Union City, CA, USA), in accordance with the manufacturer’s stated guidelines.

### 4.6. Statistical Analyses

All physiological and biochemical data were analyzed through a single-factor analysis of variance with the GLM program of SAS 8.1 statistical software (SAS Institute, Inc., Cary, NC, USA) and GraphPad Prism version 8 (GraphPad Software, La Jolla, CA, USA) software. A standard t-test or multiple t-test with Holm–Sidak multiple comparison corrections was used to compare two groups for significance. The normality test for this data was used as the Shapiro–Wilk test. One-way analysis of variance (ANOVA) with post-hoc Tukey’s multiple comparison test was used to compare the means of more than two groups. All data are presented as mean ± standard error of the mean (SEM). Differences among treatments were considered significant at *p* < 0.05 and a trend at 0.05 < *p* < 0.10.

## 5. Conclusions

In summary, dietary (3.9% reduction in crude protein level) supplementation of diets with SB, MCT, and n-3 PUFA could not compensate for the effects of the LP diet on the growth of weaned piglets. Nevertheless, it did demonstrate beneficial effects on immunity, antioxidant capacity, and bacterial community in piglets, pointing to the important role of SB, MCT, or n-3 PUFA in piglet health in cases of LP diets.

## Figures and Tables

**Figure 1 ijms-24-17592-f001:**
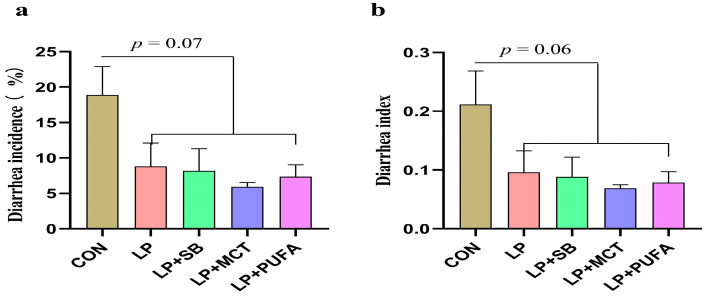
Effects of supplementing LP diets with SB, MCT, or n-3 PUFA on diarrhea rate and diarrhea index in weaned piglets. n = 24 replicates per treatment. CON, a basal diet; LP, 16.5% crude protein; LP + SB, LP with 0.2% sodium butyrate; LP + MCT, LP with 0.2% medium chain fatty acids; LP + PUFA, LP with 0.2% n-3 polyunsaturated fatty acids. (**a**) Diarrhea incidence; (**b**) diarrhea index. The results are shown as mean ± SD.

**Figure 2 ijms-24-17592-f002:**
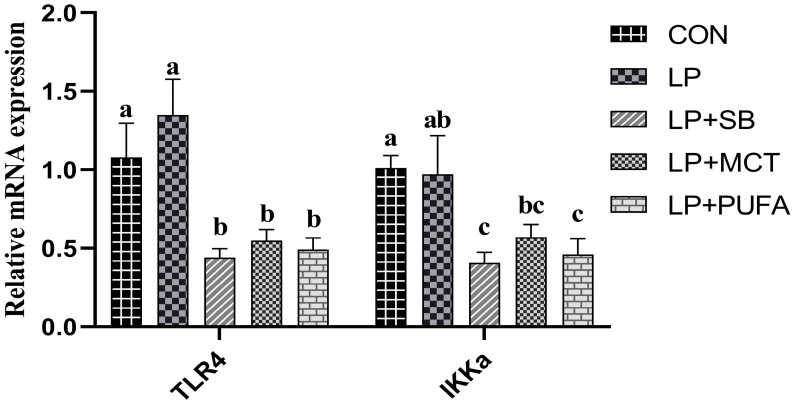
Effects of supplementing LP diets with SB, MCT, or n-3 PUFA on the relative mRNA expression of the *TLR4-IKKa* signal pathway-related genes in the colon mucosa of weaned piglets. n = 4 replicates per treatment. Abbreviations: CON, a basal diet; LP, 16.5% crude protein diet; LP + SB, LP with 0.2% sodium butyrate; LP + MCT, LP with 0.2% medium chain fatty acids; LP + PUFA, LP with 0.2% n-3 polyunsaturated fatty acids; *TLR4*, toll-like receptor 4; *IKKa*, IKK-alpha. a,b,c mean values with different superscript letters across rows are significantly different (*p* < 0.05).

**Figure 3 ijms-24-17592-f003:**
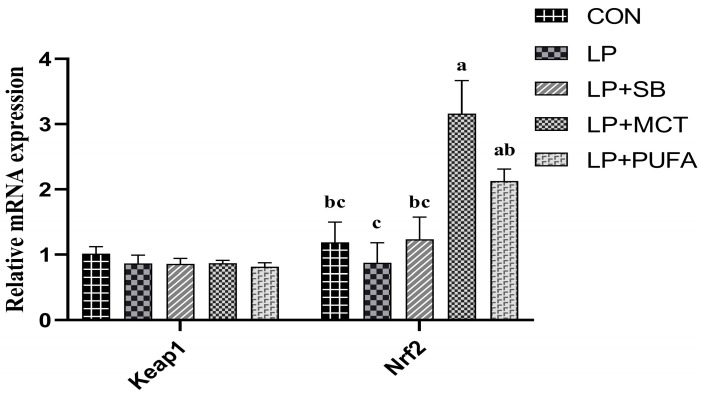
Effects of supplementing LP diets with SB, MCT, or n-3 PUFA on the relative mRNA expression of the *Keap1-Nrf2* signal pathway-related genes in the colon mucosa of weaned piglets. n = 4 replicates per treatment. Abbreviations: CON, a basal diet; LP, 16.5% crude protein diet; LP + SB, LP with 0.2% sodium butyrate; LP + MCT, LP with 0.2% medium chain fatty acids; LP + PUFA, LP with 0.2% n-3 polyunsaturated fatty acids; *Keap1*, Kelch-Like-ECH; *Nrf2*, nuclear factor erythroid 2-related facter 2. a,b,c mean values with different superscript letters across rows are significantly different (*p* < 0.05).

**Figure 4 ijms-24-17592-f004:**
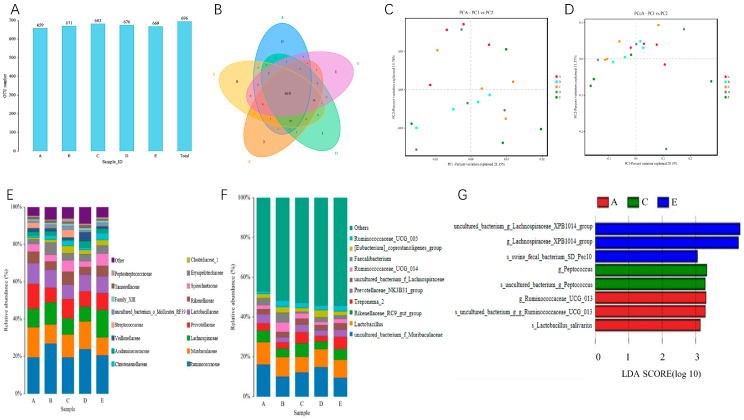
Overview of colon microbial composition in low-protein diets supplemented with SB, MCT, or n-3 PUFA. n = 6 replicates per treatment. (**A**): Distribution diagram of OTU number of the colonic microbiome. (**B**): Venn diagram at OTU level in colon. (**C**): Analysis of PCA in piglet colon. (**D**): Analysis of PCoA in piglet colon. (**E**,**F**): The relative abundance of the colon microbiome of piglets at the family level and genus level. (**G**): Microbial marker species of piglet colon. A, CON group; B, LP group; C, LP + SB group; D, LP + MCT group; E, LP + PUFA group. OUT, Each OTU corresponds to a different microbial species.

**Table 1 ijms-24-17592-t001:** Effects of supplementing low-protein diets with SB, MCT, or n-3 PUFA on the growth performance and diarrhea of weaned piglets ^1^.

Items	Treatments ^2^					SEM	*p*-Value
	CON	LP	LP + SB	LP + MCT	LP + PUFA		
Initial BW (kg)	7.93	7.93	7.93	7.93	7.93	0.05	1.00
Final BW (kg)	16.9 ^a^	14.1 ^b^	13.7 ^b^	13.3 ^b^	13.5 ^b^	0.27	<0.01
ADG, g/d	332 ^a^	229 ^b^	215 ^b^	199 ^b^	205 ^b^	10.31	<0.01
ADFI, g/d	523 ^a^	496 ^c^	517 ^ab^	497 ^c^	499 ^bc^	6.72	0.02
F/G, g/g	1.58 ^a^	2.18 ^b^	2.43 ^b^	2.54 ^b^	2.48 ^b^	0.11	<0.01

^a,b,c^ Mean values with different superscript letters across rows are significantly different (*p* < 0.05); ^1^ n = 24 replicates per treatment. ^2^ CON, a basal diet; LP, 16.5% crude protein; LP + SB, LP with 0.2% sodium butyrate; LP + MCT, LP with 0.2% medium chain fatty acids; LP + PUFA, LP with 0.2% n-3 polyunsaturated fatty acids. Abbreviations: BW, body weight; ADG, average daily gain; ADFI, average daily feed intake; F: G, feed to gain ratio.

**Table 2 ijms-24-17592-t002:** Effect of low-protein diets supplemented with SB, MCT, or n-3 PUFA on the plasma cytokines of weaned piglets ^1^.

Items	Treatments ^2^					SEM	*p*-Value
	CON	LP	LP + SB	LP + MCT	LP + PUFA		
Ig A (pg/mL)	68.0	51.8	40.7	46.0	56.6	17.6	0.85
Ig G (pg/mL)	4.17	1.79	1.45	2.32	3.60	1.18	0.43
Ig M (pg/mL)	0.98	0.72	0.69	0.70	0.77	0.27	0.94
IL-6 (pg/mL)	26.2	12.4	12.8	12.8	10.6	3.85	0.07
IL-8 (pg/mL)	2.23 ^ab^	2.45 ^a^	1.14 ^bc^	1.25 ^bc^	0.85 ^c^	0.38	0.03
IL-10 (pg/mL)	1.13	2.41	2.29	2.36	5.65	1.17	0.13
IL-17 (pg/mL)	0.28	0.39	0.32	0.20	0.74	0.15	0.15
TGF-β (pg/mL)	22.2	20.7	21.4	20.8	20.4	1.04	0.74
IFN-γ (pg/mL)	2.13	1.19	1.23	0.92	0.73	0.35	0.09
Lymphocytes goblet cell amount	60.8 ^ab^	50.6 ^b^	40.2 ^b^	86.6 ^a^	50.6 ^b^	9.35	0.02

^a,b,c^ Mean values with different superscript letters across rows are significantly different (*p* < 0.05).; ^1^ n = 6 replicates per treatment. ^2^ CON, a basal diet; LP, 16.5% crude protein; LP + SB, LP with 0.2% sodium butyrate; LP + MCT, LP with 0.2% medium chain fatty acids; LP + PUFA, LP with 0.2% n-3 polyunsaturated fatty acids.

**Table 3 ijms-24-17592-t003:** Effect of low-protein diets supplemented with SB, MCT, or n-3 PUFA on the plasma antioxidant capacity of weaned piglets ^1^.

Items	Treatments ^2^					SEM	*p*-Value
	CON	LP	LP + SB	LP + MCT	LP + PUFA		
T-AOC (mM)	1.59	1.56	1.56	1.53	1.54	0.03	0.42
CAT (U/mL)	4.27	2.29	3.50	4.97	4.18	0.67	0.10
T-SOD (U/mL)	37.2	37.9	38.6	26.6	28.0	4.67	0.21
GSH-Px (U/mL)	522 ^c^	645 ^c^	1014 ^b^	1351 ^a^	1281 ^ab^	102	<0.001
TNOS (U/mL)	16.2	17.0	16.0	16.5	16.7	1.04	0.96
iNOS (U/mL)	4.30	5.40	4.59	5.01	4.59	0.65	0.78

^a,b,c^ Mean values with different superscript letters across rows are significantly different (*p* < 0.05); ^1^ n = 6 replicates per treatment. ^2^ CON, a basal diet; LP, 16.5% crude protein; LP + SB, LP with 0.2% sodium butyrate; LP + MCT, LP with 0.2% medium chain fatty acids; LP + PUFA, LP with 0.2% n-3 polyunsaturated fatty acids.

**Table 4 ijms-24-17592-t004:** Effects of supplementing low-protein diets with SB, MCT, or n-3 PUFA on the colonic microbial alpha diversity of weaned piglets ^1^.

Items	Treatments ^2^					SEM	*p*-Value
	CON	LP	LP + SB	LP + MCT	LP + PUFA		
ACE	560	577	581	542	527	21.0	0.36
Chao1	563	576	585	556	532	21.6	0.50
Simpson	0.97	0.98	0.98	0.97	0.96	0.004	0.14
Shannon	6.61	6.81	6.85	6.54	6.35	0.14	0.13

^1^ n = 6 replicates per treatment. ^2^ CON, a basal diet; LP, 16.5% crude protein; LP + SB, LP with 0.2% sodium butyrate; LP + MCT, LP with 0.2% medium chain fatty acids; LP + PUFA, LP with 0.2% n-3 polyunsaturated fatty acids. ACE, Chao1 reflects community richness. Shannon reflects community diversity.

**Table 5 ijms-24-17592-t005:** Effects of supplementing low-protein diets with SB, MCT, or n-3 PUFA on the colonic microbial abundance of the weaned piglet at the family level ^1^.

Items (%)	Treatments ^2^					SEM	*p*-Value
	CON	LP	LP + SB	LP + MCT	LP + PUFA		
*Ruminococcaceae*	0.193	0.268	0.194	0.238	0.207	0.027	0.27
*Muribaculaceae*	0.162	0.103	0.122	0.15	0.095	0.027	0.37
*Lachnospiraceae*	0.099	0.118	0.088	0.077	0.149	0.024	0.28
*Prevotellaceae*	0.132	0.079	0.104	0.085	0.091	0.018	0.29
*Lactobacillaceae*	0.113	0.098	0.077	0.091	0.088	0.029	0.93
*Rikenellaceae*	0.061	0.044	0.069	0.041	0.055	0.016	0.73
*Spirochaetaceae*	0.042	0.037	0.06	0.032	0.064	0.021	0.74
*Erysipelotrichaceae*	0.025 ^b^	0.064 ^a^	0.036 ^b^	0.038 ^b^	0.043 ^b^	0.007	0.01
*Clostridiaceae-1*	0.009	0.016	0.04	0.023	0.03	0.01	0.30
*Christensenellaceae*	0.013	0.023	0.027	0.013	0.037	0.008	0.21
*Acidaminococcaceae*	0.019	0.021	0.006	0.034	0.025	0.014	0.76
*Veillonellaceae*	0.018	0.015	0.012	0.046	0.011	0.015	0.48
*Streptococcaceae*	0.03	0.004	0.044	0.001	0.002	0.021	0.49
*Uncultured bacterium-o-* *Mollicutes RF39*	0.009	0.016	0.02	0.01	0.017	0.004	0.29
*Family XIII*	0.01	0.011	0.013	0.014	0.013	0.003	0.90
*Peptostreptococcaceae*	0.007	0.006	0.018	0.008	0.011	0.004	0.34
*Tannerellaceae*	0.012	0.011	0.008	0.014	0.006	0.005	0.86

^a,b^ Mean values with different superscript letters across rows are significantly different (*p* < 0.05).; ^1^ n = 6 replicates per treatment. ^2^ CON, a basal diet; LP, 16.5% crude protein; LP + SB, LP with 0.2% sodium butyrate; LP + MCT, LP with 0.2% medium chain fatty acids; LP + PUFA, LP with 0.2% n-3 polyunsaturated fatty acids.

**Table 6 ijms-24-17592-t006:** Effects of supplementing low-protein diets with SB, MCT, or n-3 PUFA on the colonic microbial abundance of the weaned piglet at the genus level ^1^.

Items (%)	Treatments ^2^					SEM	*p*-Value
	CON	LP	LP + SB	LP + MCT	LP + PUFA		
*Uncultured bacterium-f-Muribaculaceae*	0.162	0.103	0.122	0.150	0.095	0.027	0.37
*Lactobacillus*	0.113	0.098	0.077	0.091	0.088	0.029	0.93
*Rikenellaceae-RC9-gut-group*	0.059	0.043	0.068	0.039	0.053	0.015	0.69
*Treponema-2*	0.038	0.033	0.055	0.030	0.060	0.020	0.75
*Prevotellaceae-NK3B31-group*	0.045	0.026	0.040	0.028	0.036	0.009	0.58
*Uncultured bacterium-f-Lachnospiraceae*	0.025	0.026	0.029	0.025	0.032	0.007	0.93
*Ruminococcaceae-UCG-014*	0.021	0.048	0.026	0.017	0.022	0.008	0.09
*Faecalibacterium*	0.035	0.047	0.008	0.022	0.018	0.016	0.47
*[Eubacterium]-coprostanoligenes_group*	0.021	0.029	0.019	0.036	0.021	0.004	0.08
*Ruminococcaceae-UCG-005*	0.013	0.033	0.027	0.023	0.029	0.007	0.39

^1^ n = 6 replicates per treatment. ^2^ CON, a basal diet; LP, 16.5% crude protein; LP + SB, LP with 0.2% sodium butyrate; LP + MCT, LP with 0.2% medium chain fatty acids; LP + PUFA, LP with 0.2% n-3 polyunsaturated fatty acids.

**Table 7 ijms-24-17592-t007:** Ingredients and chemical composition of the basal diets (air-dry basis).

Items	Treatments ^1^				
	CON	LP	LP + SB	LP + MCT	LP + PUFA
Ingredients, %					
Corn	61.45	69.37	69.23	69.23	69.23
Soybean meal	12.90	8.30	8.28	8.28	8.28
Puffed soybean	12.14	8.20	8.18	8.18	8.18
Fish meal	4.80	4.70	4.69	4.69	4.69
Soybean oil	1.80	1.90	1.90	1.90	1.90
Whey powder	2.50	2.50	2.50	2.50	2.50
CaHPO_4_	1.10	1.24	1.24	1.24	1.24
Limestone powder	0.80	0.80	0.80	0.80	0.80
Salt	0.30	0.30	0.30	0.30	0.30
L-lysine HCl	0.66	0.91	0.91	0.91	0.91
Methionine	0.25	0.32	0.32	0.32	0.32
Tryptophan	0.05	0.09	0.09	0.09	0.09
Threonine	0.25	0.37	0.37	0.37	0.37
Premix ^2^	1.00	1.00	1.00	1.00	1.00
SB	0.00	0.00	0.20	0.00	0.00
MCT	0.00	0.00	0.00	0.20	0.00
n-3 PUFA	0.00	0.00	0.00	0.00	0.20
Total	100.00	100.00	100.00	100.00	100.00
Chemical composition ^3^					
Digestive energy DE (MJ/kg)	14.67	14.58	14.59	14.59	14.60
Crude protein	20.45	16.75	16.54	16.68	16.15
Ca	0.80	0.89	0.88	0.87	0.82
Total P	0.69	0.74	0.74.	0.77	0.73
Lysine	1.56	1.55	1.55	1.55	1.55
Methionine	0.58	0.61	0.61	0.61	0.61
Cystine	0.31	0.27	0.26	0.26	0.26
Threonine	0.96	0.96	0.96	0.96	0.96
Tryptophan	0.26	0.26	0.26	0.26	0.26
Arginine	1.15	0.92	0.92	0.92	0.92
Histidine	0.45	0.39	0.39	0.39	0.39
Isoleucine	0.72	0.60	0.60	0.60	0.60
Leucine	1.60	1.41	1.40	1.40	1.40
Phenylalanine	0.81	0.68	0.67	0.67	0.67
Tyrosine	0.56	0.49	0.49	0.49	0.49
Valine	0.83	0.70	0.70	0.70	0.70

^1^ CON, a basal diet; LP, 16.5% crude protein; LP + SB, LP with 0.2% sodium butyrate; LP + MCT, LP with 0.2% medium chain fatty acids; LP + PUFA, LP with 0.2% n-3 polyunsaturated fatty acids. ^2^ Premixes provided per kg diet: VA 15,750 IU, VD_3_ 2450 IU, VE 17.5 IU, VK_3_ 1.8 mg, VB_1_ 2.1 mg, VB_2_ 7.0 mg, VB_6_ 0.4 mg, VB_12_ 0.028 mg, folic acid 0.4 mg, biotin 0.1 mg, nicotinamide 28.0 mg, D-pantothenic acid 15.8 mg, zinc as zinc sulfate 170 mg, ferrous as ferrous sulfate 140 mg, manganese as manganese sulfate 34 mg, copper as copper sulfate 16 mg, iodine as potassium iodide 0.55 mg, selenium as sodium selenite 0.29 mg. ^3^ The crude protein, calcium, and phosphorus are measured values.

**Table 8 ijms-24-17592-t008:** The sequences of primers.

Genes ^1^	Primer Sequences (5′→3′) Primer Sequence (5′→3′)	Serial Number	Tm Value (°C)	Length of PCR (bp)
*TLR4*	F: CTGCCTGTGCTGAGTTTCAGGAACG R: CCTCACCCAGTCTTCGTC	NM-001293316.1	66 57	219
*IKKα*	F: AATCTGCTTCGGAACAACA R: GTCAATCTGGATGCTGGTT	XM-021077172.1	55 55	111
*Keap1*	F: AGCAGCGGCGTTTCTACGT R: TGGGCTTGTGCAGAGTGAGC	XM_021076667.1	63 63	168
*Nrf2*	F: TCAGACCCACCACTAGCCTT R: GTGATGCCAGCAGACCTCTT	XM_021075133.1	50 50	138
*β-actin*	F: TGCGGCATCCACCAAACTA R: CGTAGAGGTCCTTGCGGATGT	XM_021086047.1	57 62	70

^1^ *TLR4*, toll-like receptor 4; *IKKa*, IKK-alpha; *Keap1*, Kelch-Like-ECH; *Nrf2*, nuclear factor erythroid 2-related factor 2; β-actin, beta-actin.

## Data Availability

Data are contained within the article.

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
