# Peer review of "Effect of Low Protein Diets Supplemented with Sodium Butyrate, Medium-Chain Fatty Acids, or n-3 Polyunsaturated Fatty Acids on the Growth Performance, Immune Function, and Microbiome of Weaned Piglets"

_ijms, 2023, doi:10.3390/ijms242417592_

Round 1
Reviewer 1 Report
Comments and Suggestions for Authors
In this study, the authors investigated the Effect of Low Protein Diets Supplemented with Sodium Butyrate, Medium-chain Fatty Acids, or n-3 Polyunsaturated Fatty Acids on the Growth Performance, Immune Function, and Microbiome of Weaned Piglets. However, there are some revisions that should be made before publication.
The majority of cited references in the manuscript need to be between brackets please revise.
When analyzing the incidence of diarrhea in Table 1, it is important to use appropriate statistical methods. Non-parametric tests such as the chi-square test or Fisher's exact test are commonly employed when dealing with categorical or frequency data. These tests are suitable for determining if there is a significant association or difference between groups. However, using ANOVA (Analysis of Variance) to analyze the incidence of diarrhea in Table 1 would be inappropriate. ANOVA is typically used to compare means between two or more continuous variables. It assumes normality of the data and equal variances across groups, which may not hold true for frequency data like diarrhea incidence. Please revise this point with a specialist in statistical analysis.
In statistical analysis part in M&M you mentioned that you used one way a nova to examine the significant effects of different treatments. However, upon reviewing the recorded results, it is evident that the general linear model (GLM) was actually used. Please revise accurately this point.
To determine the normality of the data, statistical tests such as the Shapiro-Wilk test, Kolmogorov-Smirnov test, or Anderson-Darling test were employed. These tests assess whether the data significantly deviate from a normal distribution. Therefore, as part of the data analysis process, the normality assumption was carefully examined using relevant statistical tests to ensure the validity of subsequent statistical analyses and interpretations.
Although the health status of the piglets improved significantly, it is worth noting that there was a significant decrease in live body weight. Considering this outcome, it would be prudent to calculate the economic efficiency of the tested diets. In addition to evaluating the health benefits of the diets, assessing their economic impact is crucial for making informed decisions. By calculating the economic efficiency, we can determine the cost-effectiveness of the diets in terms of piglet growth and weight gain. This analysis takes into account not only the improvements in health but also the potential economic losses associated with a decrease in live body weight. By incorporating economic factors into the assessment, we can obtain a more comprehensive understanding of the overall effectiveness of the tested diets. This information will be valuable for determining the feasibility and profitability of implementing these diets in practical settings.
Please adjust the resolution of Figure 3
Author Response
Please see in the attachment

Reviewer 2 Report
Comments and Suggestions for Authors
Effect of Low Protein Diets Supplemented with Sodium Butyr- 2 ate, Medium-chain Fatty Acids, or n-3 Polyunsaturated Fatty 3 Acids on the Growth Performance, Immune Function, and Mi- 4 crobiome of Weaned Piglets
The paper evaluates the effect of diets low in protein level and the combination of SB, MCFA and n-3 fatty acids. It is an interesting work, but it requires of improvements. Some specific comments are listed below. The material and methods need to be clarified. Also, some aspects of the results need to be checked because information presented in tables does not fit with what the authors write in the paper. Also some sentences should be rewritten in the discussion section.
Specific comments.
Lines 47, 49… 7 as reference should be included in brackets. Same for 8. Check all text.
Line 63 “We speculated that supplementing 62 LP diets with SB, MCT, or n-3 PUFA have would beneficial effects..”
Have would or would have??
Line 69, “The ADFI of piglets fed LP, LP+SB, LP+MCT, or LP+PUFA is significantly lower than that 69 of piglets fed the CON diet (P < 0.05)”
“was” instead “is”. Sometimes you write in present and others in past form.
Line 98-99 “and the plasma GSH-Px activity of piglets in the LP + SB group 98 was significantly higher than that in LP + MCT group (P < 0.05)”
But according to table 3, GSH-px was higher in LP+MCT than in LP+SB group
Line 225. The regulation of im-munity by MCFA is influenced by the MCFA.
What do you mean in this sentence??
Line 229. “The results of this study found that the LP group had 229 higher levels of IL-10 than the CON group, while the LP+PUFA group tended to have 230 greater levels of IL-10 than the CON group”
But according to table differenteces in IL-10 were not statistically significant. Rewrite this paragraph
Line 251-253. In agreement with previous findings, SB increased CAT and T-SOD con- 251 tent 28, and capric acid was able to increase T-SOD and G SH-Px mRNA expression in 252 porcine epithelial cells 29.
But in the presentnstudy SOD and CAT were not affected. Delete”in agreement”. Rewrite.
Lines 264-277. It would be easier to follow for authors to explain functions of TLR and iKKa
Line 282. What is OUT, and ACE, Chao 1, Shannon…Clarify all the informartion presented in this paragraph.
Line 327. What kind of SB was used??. 98% purity, then all butiryc acid? So how much was the part of salt? Protected?. Specify more information.
Line 329 The primary active ingredients 329 of the n-3 PUFA are alpha-linolenic acid, docosahexaenoic acid, and eicosapentaenoic acid.
Change “are” by “were”. If purity of n-3 was 50 %, what was the other 50%??
And what about MCFA. What was the composition?
Lines 345-346. The weight of piglets was recorded on days 1 and 28, and feed consumption was 345 recorded daily. The average daily gain (ADG), average daily feed intake (ADFI), and (G: 346 F; gain to feed ratio) were calculated from days 1 to 28
If you used weaned piglets at 28 days why did you weight on day 1 and 28. The initial time for the experiment is day 28 as you stated in line 322 “28-day-old weaned piglets”. So when was the second time that you register piglets weights?. Please clarify.
And again in line 356 you say “on the 29th day, six piglets from each 356 treatment group were chosen at random for slaughter”. So day 29 was the end of the experimento or the start of the experiment?
Line 359. The plasma was extracted by centrifugation at 3,000 × g for 359 15 minutes. Next, piglets were anesthetized with an intravenous injection of sodium pen- 360 tobarbital (60 mg/kg BW), and the pigs were humanely slaughtered
Please indicate the number of ethical autorization and ethical statements in any place of the paper.
Comments on the Quality of English LanguageMinor editing English required
Author Response
Dear Reviewer
Thank you for your comments concerning our manuscript, These comments are valuable and very helpful for revising and improving our paper, as well as the important guiding significance to our researches. We have studied comments carefully and have made the revisions that are highlighted in yellow in the manuscript, and we hope these revisions can meet with approval. Our responses are as flowing:
Point 1: Lines 47, 49… 7 as reference should be included in brackets. Same for 8. Check all text.
Response 1: Thank you for carefully and patiently reviewing our manuscript and listing mistakes in the original manuscript. I have checked all references and corrected them. (Lines 56~58)
Point 2: Line 63 “We speculated that supplementing 62 LP diets with SB, MCT, or n-3 PUFA have would beneficial effects..”Have would or would have??
Response 2: I think it should be “would have”. (Line 72)
Point 3: Line 69, “The ADFI of piglets fed LP, LP+SB, LP+MCT, or LP+PUFA is significantly lower than that 69 of piglets fed the CON diet (P < 0.05)”. “was” instead “is”. Sometimes you write in present and others in past form.
Response 3: I think it should be “was”.
Point 4: Line 98-99 “and the plasma GSH-Px activity of piglets in the LP + SB group 98 was significantly higher than that in LP + MCT group (P < 0.05)”. But according to table 3, GSH-px was higher in LP+MCT than in LP+SB group.
Response 4: Compared to the CON and LP groups, and the plasma GSH-Px activity of piglets in the LP + MCT group was significantly higher than that in LP + SB group (p < 0.05). (Lines 115~117)
Point 5: Line 225. The regulation of im-munity by MCFA is influenced by the MCFA. What do you mean in this sentence??
Response 5: The regulation of immunity by MCFA is influenced by the type of MCFA of MCFA, in an in vitro study, octanoic acid inhibited IL-8 production by Caco-2 cells, while decanoic acid promoted IL-8 secretion. (Lines 244~247)
Point 6: Line 229. “The results of this study found that the LP group had 229 higher levels of IL-10 than the CON group, while the LP+PUFA group tended to have 230 greater levels of IL-10 than the CON group”. But according to table differenteces in IL-10 were not statistically significant. Rewrite this paragraph
Response 6: The MCT used in this study was a mixture of octanoic and decanoic acid, and MCFA was perhaps more effective in immune effects. The results of this study found that there were no significant differences in plasma IL-10 in piglets between the five groups. Inconsistency with previous study, feeding LP rations significantly increased IL-10. Related study has found that both SB and n-3 PUFA enhance the production of IL-10 anti-inflammatory factors. The effect of PUFA in promoting IL-10 was more pronounced in the conclusion of the present study. The promotion of IL-10 by LP diets may manifest itself as the experiment of this study continues to be extended. (Lines 247~254)
Point 7: Line 251-253. In agreement with previous findings, SB increased CAT and T-SOD con- 251 tent 28, and capric acid was able to increase T-SOD and G SH-Px mRNA expression in 252 porcine epithelial cells 29. But in the presentnstudy SOD and CAT were not affected. Delete”in agreement”. Rewrite.
Response 7: Inconsistent with previous findings, SB increased CAT and T-SOD content. The possible reason for this is that SB was added at an insufficient level, and further research is needed to determine this. Capric acid was able to increase T-SOD and GSH-Px mRNA expression in porcine epithelial cells. (Lines 269~272)
Point 8: Lines 264-277. It would be easier to follow for authors to explain functions of TLR and iKKa.
Response 8: TLR4 recognizes Gram-negative bacterial lipopolysaccharide (LPS) and also heat-shockproteins (HSP) released by host necrotic cells. TLR4 mainly produces IL-12 p70, IFN-γ mediating protein (IP-10) and transcriptional IFN-β. Under the stimulation of TNF-α, IKKα can directly enter the nucleus and activate the expression of specific NF-κB-responsive genes by catalyzing the phosphorylation of histone H3.
Point 9: Line 282. What is OUT, and ACE, Chao 1, Shannon…Clarify all the informartion presented in this paragraph.
Response 9: OUT: Each OTU corresponds to a different microbial species. ACE, Chao1 reflects community richness. Shannon reflects community diversity. (Line 210, and line 190)
Point 10: Line 327. What kind of SB was used??. 98% purity, then all butiryc acid? So how much was the part of salt? Protected?. Specify more information.
Response 10: The molecular formula of sodium butyrate (SB) was C4H7O2Na, then all sodium butyrate. SB is protected. The part of salt was 2%.
Point 11: Line 329 The primary active ingredients 329 of the n-3 PUFA are alpha-linolenic acid, docosahexaenoic acid, and eicosapentaenoic acid. Change “are” by “were”. If purity of n-3 was 50 %, what was the other 50%??
Response 11: I've changed it to "were". The other 50% is fish oil.
Point 12: And what about MCFA. What was the composition?
Response 12: The main components of MCFA are caprylic and capric acids, with the remaining 50% being medium chain triglycerides.
Point 13: Lines 345-346. The weight of piglets was recorded on days 1 and 28, and feed consumption was 345 recorded daily. The average daily gain (ADG), average daily feed intake (ADFI), and (G: 346 F; gain to feed ratio) were calculated from days 1 to 28. If you used weaned piglets at 28 days why did you weight on day 1 and 28. The initial time for the experiment is day 28 as you stated in line 322 “28-day-old weaned piglets”. So when was the second time that you register piglets weights?. Please clarify.
Response 13: The second time piglet weights were recorded was at 8:00 a.m. on day 29. (Line 365)
Point 14: And again in line 356 you say “on the 29th day, six piglets from each 356 treatment group were chosen at random for slaughter”. So day 29 was the end of the experimento or the start of the experiment?
Response 14: Day 29 at 8 a.m. was the end of the experiment.
Point 15: Line 359. The plasma was extracted by centrifugation at 3,000 × g for 359 15 minutes. Next, piglets were anesthetized with an intravenous injection of sodium pen- 360 tobarbital (60 mg/kg BW), and the pigs were humanely slaughtered. Please indicate the number of ethical autorization and ethical statements in any place of the paper.
Response 15: Institutional Review Board Statement: The protocol was approved by the experimentation rules of the Southwest University Animal ethics and Use Committee (ethical license number: IACUC-20210731-01). The number of ethical autorization is 120. (Lines 454~456)
Reviewer 3 Report
Comments and Suggestions for Authors
The manuscript entitled "Effect of Low Protein Diets Supplemented with Sodium Butyrate, Medium-chain Fatty Acids, or n-3 Polyunsaturated Fatty Acids on the Growth Performance, Immune Function, and Microbiome of Weaned Piglets" is well written. But, it needs minor revision. Please refer to comments given in the text of reviewed attached file of the manuscript.

Author Response
Dear Reviewer
Thank you for your comments concerning our manuscript, These comments are valuable and very helpful for revising and improving our paper, as well as the important guiding significance to our researches. We have studied comments carefully and have made the revisions that are highlighted in yellow in the manuscript, and we hope these revisions can meet with approval. Our responses are as flowing:
Point 1: This conclusion is wide and general. It is better to add your specific conclusion from your specific results.
Response 1: Thank you for carefully and patiently reviewing our manuscript and listing mistakes in the original manuscript. In conclusion, this study showed that LP diets supplemented with SB, MCT, or n-3 PUFA reduced plasma inflammatory factor levels, increased plasma GSH-Px activity, and declined mRNA expression of TLR4 and IKKa in the colonic epithelium, whereas reduced the abundance of Erysipelotrichaceae in the colon of piglets. (Lines 32~35)
Point 2: It is better to explain about importance of growth and production in farm animals. For this, you can sue below added sentences and references:
Response 2: Thank you, I have added. (Lines 46~49)
Point 3: What is the superiority of your research compared to other researches?
Response 3: This study aimed to investigate the effects of low-protein (LP) diets supplemented with sodium butyrate (SB), medium-chain fatty acids (MCT), or n-3 polyunsaturated fatty acids (n-3 PUFA) on the growth performance, immune function, and the microbiome of weaned piglets. The main objective is to utilize the advantages of low-protein diets and fatty acids applied to piglets, and to provide a theoretical basis for the further development of low-protein feed technology.
Point 4: please identify sex of used animals.
Response 4: Split evenly between males and females. (Line 343)
Point 5: Please add company and country for used kits and devices.
Response 5: Code, R0016. Beyotime Biotechnology, Shanghai, China.
Point 6: please explain how did you design the primers and which software did you use for designing.
Response 6: The primers were designed by Primer Premier 5 software, and the primer sequences were verified in NCBI GenBank.
1, Primers are specific within the conserved region of the DNA sequence.
2, The primer length is generally between 15-30 bases.
3, GC content between 40% -60%, the amplified fragment is also the best GC content between 40% -60%, so that it is easier to amplify; primer Tm value range of 55-66 ℃, upstream and downstream primer Tm value should not be too big a difference, it is best not to exceed 5 degrees. 4, the primer 3 ′ end can not be chosen A, it is best to choose T.
5, The bases should be randomly distributed, especially the 3′ end should not be more than 3 consecutive G or C.
6, There should be no complementary sequences between the primers themselves and the primers to prevent the formation of a hairpin structure (Hairpin).
Point 7: Table 2 or 8?
Response 7: It is Table 8. (Line 406)
Point 8: please add fragment size of PCR products for each primer in the table.
Response 8: (Lines 414~416)
|
Genes1 |
Primer sequences (5′→3′) Primer sequence (5′→3′) |
Serial number |
Tm value (℃) |
Length of PCR (bp) |
|
TLR4 |
F: CTGCCTGTGCTGAGTTTCAGGAACG R: CCTCACCCAGTCTTCGTC |
NM-001293316.1 |
66 57 |
219 |
|
IKKα |
F: AATCTGCTTCGGAACAACA R: GTCAATCTGGATGCTGGTT |
XM-021077172.1 |
55 55 |
111 |
|
Keap1 |
F: AGCAGCGGCGTTTCTACGT R: TGGGCTTGTGCAGAGTGAGC |
XM_021076667.1 |
63 63 |
168 |
|
Nrf2 |
F: TCAGACCCACCACTAGCCTT R: GTGATGCCAGCAGACCTCTT |
XM_021075133.1 |
50 50 |
138 |
|
β-actin |
F: TGCGGCATCCACCAAACTA R: CGTAGAGGTCCTTGCGGATGT |
XM_021086047.1 |
57 62 |
70 |
Point 9: Please add used statistical animal model and its components in the text of the manuscript.
Response 9: All physiological and biochemical data were analyzed through a single-factor analysis of variance with the GLM program of SAS 8.1 statistical software (SAS Institute, Inc., Cary, NC, USA) and GraphPad Prism version 8 (GraphPad Software, La Jolla, CA, USA) software. A standard t-test or multiple t-test with Holm–Sidak multiple comparison corrections was used to compare two groups for significance. One-way analysis of variance (ANOVA) with post-hoc Tukey’s multiple comparison test was used to compare the means of more than two groups. All data are presented as mean ± standard error of the mean (SEM). Differences among treatments were considered significant at P < 0.05 and a trend at 0.05 < P < 0.10. (Lines 431~439)
Round 2
Reviewer 1 Report
Comments and Suggestions for Authors
Thank you for your effort but i still have some comments
Mann-Whitney U Test is used if there are only two groups. But in this case you should use Kruskal-Wallis test and obtain that in statistical analysis part in M&M section.
You stated that you used A standard t-test or multiple t-test with Holm–Sidak multiple comparison corrections to compare two groups for significance but I mean the testes for examining data normality. Please revise again.
Author Response
Dear Reviewer,
Thank you for your valuable comments on our manuscript again, which are very helpful for us to revise and improve the paper, and also have important guidance for our research. We have studied comments carefully and have made the revisions that are highlighted in yellow in the manuscript, and we hope these revisions can meet with approval. Our responses are as flowing:
Point 1: Mann-Whitney U Test is used if there are only two groups. But in this case you should use Kruskal-Wallis test and obtain that in statistical analysis part in M&M section.
Response 1: First of all, thank you for once again carefully and patiently reviewing our manuscript and listing mistakes in the original manuscript. Diarrhea rate and diarrhea index were tested using nonparametric tests and the results were used Kruskal-Wallis test. I have presented in the article. (Lines 92~97)
Point 2: You stated that you used A standard t-test or multiple t-test with Holm–Sidak multiple comparison corrections to compare two groups for significance but I mean the testes for examining data normality. Please revise again.
Response 2: Since the sample size of the data in this study was less than 50, the normality test for this data was used as the Shapro-Wilk test. I have presented in the original manuscript. (Lines 454~455)
Reviewer 2 Report
Comments and Suggestions for Authors
The authors have improved the paper and have answered many of the questions. However, some of the information that the authors provide in the answers have not been included in the paper. This is mainly important in the material and methods section. Moreover, authors must clarify the initial and final times of the experiment, since in my opinion there are some mistakes in the text. The start seems to be on day 28 after weaning with aprox 7 kg but the end authors answered day 29 with 14 Kg??. But they calculate performances from 1-28 days.
Response 7: Inconsistent with previous findings, SB increased CAT and T-SOD content. The possible reason for this is that SB was added at an insufficient level, and further research is needed to determine this. Capric acid was able to increase T-SOD and GSH-Px mRNA expression in porcine epithelial cells. (Lines 269~272)
Lines 266-268, Inconsistent??. I think is better to delete “inconsistent”. A possibility could be “other authors reported SB increased…… The possible reason of this result could be that SB was added at an insufficient level to be combined with LP diets, so further research is needed.
Response 8: TLR4 recognizes Gram-negative bacterial lipopolysaccharide (LPS) and also heat-shockproteins (HSP) released by host necrotic cells. TLR4 mainly produces IL-12 p70, IFN-γ mediating protein (IP-10) and transcriptional IFN-β. Under the stimulation of TNF-α, IKKα can directly enter the nucleus and activate the expression of specific NF-κB-responsive genes by catalyzing the phosphorylation of histone H3.
Please include some of this information in the paper when you discuss each specific data, to identify the important of TLR4 and IKKa
Point 10: Line 327. What kind of SB was used??. 98% purity, then all butiryc acid? So how much was the part of salt? Protected?. Specify more information.
Please include this information in material and methods section when talking about the diets, line 347
Response 10: The molecular formula of sodium butyrate (SB) was C4H7O2Na, then all sodium butyrate. SB is protected. The part of salt was 2%.
Please include that SB was protected in material and methods section when talking about the diets, line 347
Response 12: The main components of MCFA are caprylic and capric acids, with the remaining 50% being medium chain triglycerides.
Please include this information in material and methods section when talking about the diets, line 347
Response 13: The second time piglet weights were recorded was at 8:00 a.m. on day 29. (Line 365)
So if you started the experiment at day 28 with weaned piglets and aprox 7 kg, the experiment finish on day 29 with aprox 14 kg?. In one day the piglets gained aprox 7 kg??. Clarify the final day of the experiment in the text (lines 362 “weight of the piglets was recorded on day 1 and 29”).
Also clarify lines 363-364 “. The average daily gain (ADG), average daily feed intake (ADFI), and (G: F; gain to feed ratio) were calculated from days 1 to 28”. According to your answer from 28 to 29 days. This can not be correct. You must clarify when the experiment start and when finish. I suppose if you are talking about weaned piglets the experiment start at day 28 but when was finished??
Also include this information in TAble 1 (initial BW day 28), final BW day X
Comments on the Quality of English Language
Minor editing
Author Response
Dear Reviewer,
Thank you for your valuable comments on our manuscript again, which are very helpful for us to revise and improve the paper, and also have important guidance for our research. We have studied comments carefully and have made the revisions that are highlighted in yellow in the manuscript, and we hope these revisions can meet with approval. Our responses are as flowing:
Point 1: Lines 266-268, Inconsistent??. I think is better to delete “inconsistent”. A possibility could be “other authors reported SB increased…… The possible reason of this result could be that SB was added at an insufficient level to be combined with LP diets, so further research is needed.
Response 1: First of all, thank you for once again carefully and patiently reviewing our manuscript and listing mistakes in the original manuscript. I have revised the issues you raised, and presented them in the original manuscript. (Lines 273~275)
Point 2: Please include some of this information in the paper when you discuss each specific data, to identify the important of TLR4 and IKKa.
Response 2: TLR4, a member of the Toll-like receptor (TLR) family, is involved in innate immunity and mediates inflammatory responses by recognizing lipopolysaccharide (LPS) or bacterial endotoxins. Under the stimulation of TNF-α, IKKα can directly enter the nucleus and activate the expression of specific NF-κB-responsive genes by catalyzing the phosphorylation of histone H3. I have identified the importance of TLR4 and IKKa in the original manuscript. (Lines 290~295)
Point 3: Line 327. What kind of SB was used??. 98% purity, then all butiryc acid? So how much was the part of salt? Protected?. Specify more information. Please include this information in material and methods section when talking about the diets, line 347
The main components of MCFA are caprylic and capric acids, with the remaining 50% being medium chain triglycerides. Please include this information in material and methods section when talking about the diets, line 347
Response 3: The SB (purity ≥ 98%), MCFA (purity ≥ 50%), or n-3 PUFA (purity ≥ 50%) were sourced from Longyan Xinao Biotechnology Co., Ltd. based in Longyan, China. The molecular formula of SB was C4H7O2Na, then all sodium butyrate. The part of salt was 2%. The main components of MCFA were caprylic and capric acids, with the remaining 50% being medium chain triglycerides. The primary active ingredients of the n-3 PUFA were alpha-linolenic acid, docosahexaenoic acid, and eicosapentaenoic acid. SB, MCFA, and PUFA were protected. I've added the SB, MCFA, and PUFA info to the original manuscript. (Lines 362~368)
Point 4: So if you started the experiment at day 28 with weaned piglets and aprox 7 kg, the experiment finish on day 29 with aprox 14 kg?. In one day the piglets gained aprox 7 kg??. Clarify the final day of the experiment in the text (lines 362 “weight of the piglets was recorded on day 1 and 29”).
Also clarify lines 363-364 “. The average daily gain (ADG), average daily feed intake (ADFI), and (G: F; gain to feed ratio) were calculated from days 1 to 28”. According to your answer from 28 to 29 days. This can not be correct. You must clarify when the experiment start and when finish. I suppose if you are talking about weaned piglets the experiment start at day 28 but when was finished??
Also include this information in TAble 1 (initial BW day 28), final BW day X
Response 4: First of all, I think you have misunderstood the meaning of this experiment. We would like to explain that the animal model used in this experiment is the 28 day weaner piglet. 28 day weaner piglets are piglets from birth to day 28. The purpose of feeding 28-day-old piglets for 3 days is to adapt to the change in feed. In this experiment, 31-day weaned piglets were continued for 28 days. This experiment started with 32 day old piglets and ended with 59 day old pigs. The first day of this experiment is not the same day as the first day of the piglet's life.
The first day of this experiment is equivalent to the 32nd day of piglet growth.
Piglet weights were recorded at 8:00 a.m. on days 1 and 29 of the experiment, and feed consumption was recorded daily.
In the end, I have added all the information to the original manuscript. (Lines 359~361), (Lines 383~386)